# Evaluation of Bi-Lateral Co-Infections and Antibiotic Resistance Rates among COVID-19 Patients in Lahore, Pakistan

**DOI:** 10.3390/medicina58070904

**Published:** 2022-07-06

**Authors:** Azka Rizvi, Muhammad Umer Saeed, Ayesha Nadeem, Asma Yaqoob, Ali A. Rabaan, Muhammed A. Bakhrebah, Abbas Al Mutair, Saad Alhumaid, Mohammed Aljeldah, Basim R. Al Shammari, Hawra Albayat, Ameen S. S. Alwashmi, Firzan Nainu, Yousef N. Alhashem, Muhammad Naveed, Naveed Ahmed

**Affiliations:** 1Department of Microbiology, Pakistan Kidney and Liver Institute & Research Center, Lahore 54000, Punjab, Pakistan; azka.rizvi@pkli.org.pk; 2Department of Medical Education, King Edward Medical University, Lahore 54000, Punjab, Pakistan; usb456cpmc@gmail.com (M.U.S.); ayeshanadeem0987@gmail.com (A.N.); 3Department of Microbiology, Fatima Memorial Medical and Dental College, Lahore 54000, Punjab, Pakistan; asma.yaqoob@fmhcmd.edu.pk; 4Molecular Diagnostic Laboratory, Johns Hopkins Aramco Healthcare, Dhahran 31311, Saudi Arabia; 5College of Medicine, Alfaisal University, Riyadh 11533, Saudi Arabia; 6Department of Public Health and Nutrition, The University of Haripur, Haripur 22610, Khyber Pakhtunkhwa, Pakistan; 7Life Science and Environment Research Institute, King Abdulaziz City for Science and Technology (KACST), Riyadh 11442, Saudi Arabia; mbakhrbh@kacst.edu.sa; 8Research Center, Almoosa Specialist Hospital, Al-Ahsa 36342, Saudi Arabia; abbas.almutair@almoosahospital.com.sa; 9College of Nursing, Princess Norah Bint Abdulrahman University, Riyadh 11564, Saudi Arabia; 10School of Nursing, Wollongong University, Wollongong, NSW 2522, Australia; 11Nursing Department, Prince Sultan Military College of Health Sciences, Dhahran 33048, Saudi Arabia; 12Administration of Pharmaceutical Care, Al-Ahsa Health Cluster, Ministry of Health, Al-Ahsa 31982, Saudi Arabia; saalhumaid@moh.gov.sa; 13Department of Clinical Laboratory Sciences, College of Applied Medical Sciences, University of Hafr Al Batin, Hafr Al Batin 39831, Saudi Arabia; mmaljeldah@uhb.edu.sa (M.A.); bralshammari@uhb.edu.sa (B.R.A.S.); 14Infectious Disease Department, King Saud Medical City, Riyadh 7790, Saudi Arabia; hhalbayat@gmail.com; 15Department of Medical Laboratories, College of Applied Medical Sciences, Qassim University, Buraydah 51452, Saudi Arabia; aswshmy@qu.edu.sa; 16Department of Pharmacy, Faculty of Pharmacy, Hasanuddin University, Makassar 90245, Indonesia; firzannainu@unhas.ac.id; 17Department of Clinical Laboratory Sciences, Mohammed AlMana College of Health Sciences, Dammam 34222, Saudi Arabia; yousefa@machs.edu.sa; 18Department of Biotechnology, Faculty of Science and Technology, University of Central Punjab, Lahore 54000, Punjab, Pakistan; dr.naveed@ucp.edu.pk; 19Department of Microbiology and Parasitology, School of Medical Sciences, Universiti Sains Malaysia, Kubang Kerian 16150, Kelantan, Malaysia

**Keywords:** *Enterobacteriaceae*, bacterial infections, antibiotic stewardship, hospital acquired infections, antibiotic resistance

## Abstract

*Background and Objective:* Bacterial infections are among the major complications of many viral respiratory tract illnesses, such as influenza and coronavirus disease-2019 (COVID-19). These bacterial co-infections are associated with an increase in morbidity and mortality rates. The current observational study was conducted at a tertiary care hospital in Lahore, Pakistan among COVID-19 patients with the status of oxygen dependency to see the prevalence of bacterial co-infections and their antibiotic susceptibility patterns. *Materials and Methods*: A total of 1251 clinical samples were collected from already diagnosed COVID-19 patients and tested for bacterial identification (cultures) and susceptibility testing (disk diffusion and minimum inhibitory concentration) using gold standard diagnostic methods. *Results*: From the total collected samples, 234 were found positive for different bacterial isolates. The most common isolated bacteria were *Escherichia coli* (*E. coli*) (*n* = 62) and *Acinetobacter baumannii* (*A. baumannii*) (*n* = 47). The *E. coli* isolates have shown the highest resistance to amoxicillin and ampicillin, while in the case of *A. baumannii*, the highest resistance was noted against tetracycline. The prevalence of methicillin resistant *Staphylococcus aureus* (MRSA) was 14.9%, carbapenem resistant *Enterobacteriaceae* (CRE) was 4.5%, and vancomycin resistant *Enterococcus* (VRE) was 3.96%. *Conclusions:* The results of the current study conclude that empiric antimicrobial treatment in critically ill COVID-19 patients may be considered if properly managed within institutional or national level antibiotic stewardship programs, because it may play a protective role in the case of bacterial co-infections, especially when a patient has other AMR risk factors, such as hospital admission within the previous six months.

## 1. Introduction

The novel coronavirus disease-2019 (COVID-19) has caused a global pandemic of severe respiratory tract illness [1,2]. The causative agent for this lethal disease is SARS-CoV-2 [3]. This disease made many hospitalized patients more critical, and COVID-19 cases become higher in a very short time period. It resulted in 445 million (M) reported cases and 6 M deaths worldwide [4]. Out of which, cases from Pakistan accounted for 1.15 M and 30,265 deaths [5]. The high transmissibility and associated mortality rates have raised concerns among clinicians and scientists [6,7].

During the incubation period, the virus is transmissible and upon infection, the virus induces viremia which causes fever, pharyngalgia, tiredness, diarrhea, and other symptoms [8,9]. This comprises the incubation period (1–14 days, usually 3–7 days) and early illness phases [10]. In the later stage, patients become hyper-coagulable and D-Dimer-based coagulation factors become aberrant [11]. Many other factors like demographics, immune status of the patient, treatment options, and secondary infections also contribute to the disease progression and prognosis of the disease, with secondary infection being at the top [12]. The use of empirical therapy in hospitalized COVID-19 patients has been reported widely in the literature leading to high antimicrobial resistance rates (AMR) among the bacterial isolates [12]. This raised a major concern among clinicians when deciding on treatment options for critical patients. The high AMR rates are another rising concern worldwide, and attempts are being made to tackle this problem [13].

In addition to the primary SARS-CoV-2 infection during the current COVID-19 pandemic, several secondary comorbidities are developing and significantly increasing the fatality rate [14]. Bacterial and fungal co-infections among them are critical, causing the death of many COVID-19 patients [5]. Pathogens that cause respiratory co-infection include viruses, bacteria, fungi, and other common pathogens [15]. Bacteria are one of the most often isolated organisms responsible for secondary co-infection. The bacterial co-infections may significantly increase the overall mortality among COVID-19 patients [4]. However, due to a lack of staff during the COVID-19 pandemic, many hospitals and healthcare facilities were unable to make all of the necessary diagnoses using the gold standard diagnostic techniques [9]. Secondary infections have been reported in 10–15% of the positive cases of COVID-19. Moreover, the incidence of these infections was higher among critically ill COVID-19 patients [4]. Of these co-infected patients, those who received invasive procedures accounted for up to 70% of the patients [14]. Secondary infections associated with COVID-19, especially in hospitalized patients, pose a great risk to the COVID patients as well as other (COVID-negative) patients, as there is a risk of spill-over of infectious bugs to the community [5]. 

Therefore, prolonged antibiotic treatment for secondary infections in COVID patients is of great public health concern [2]. The present retrospective observational study was designed to study the appearance of secondary bacterial infections associated with COVID-19 and their antimicrobial susceptibility patterns among patients with no oxygen support, oxygen support, and on ventilator, so that an informed decision can be made in choosing the appropriate treatment and patient management strategies. The purpose of the study was to help clinicians in the empirical therapy and infection control department of hospitals manage the spread of infectious bugs.

## 2. Materials and Methods

### 2.1. Sample Collection

Samples included in the current study were processed after the request from a clinician suspecting a secondary infection among hospitalized COVID-19 patients based on their signs and symptoms during their illness or treatment course, and not necessarily upon admission. The COVID-19 status of the patients was confirmed using nasopharyngeal swabs-based polymerase chain reaction (PCR) tests before the admission of patients in COVID-19 allocated wards. Samples were collected aseptically in sterile containers from a tertiary care hospital in Lahore, Pakistan during the period from 24th July 2021 to 31st October 2021. Samples were collected by following the standard protocol for COVID-19 infection and transferred to a microbiology lab under refrigeration conditions for microbiological testing. Pre-defined criteria and a data collection sheet were established which included the patient’s demographic details, and comorbidities.

### 2.2. Isolation and Identification of Bacterial Isolates

All of the clinical samples (except urine samples) were inoculated on a set of culture media plates (blood agar, chocolate agar, and MacConkey agar plates) and incubated at 37 °C. The urine samples were inoculated on cysteine lactose electrolyte deficient (CLED) agar. Gram smears were prepared for all specimens except urine samples, while the wet smear examination was performed for urine samples to observe the presence of pus cells. The Gram stain smears of sputum samples were observed for the presence of pus cells counted to support the assumption of possible secondary bacterial infection in combination with signs and symptoms of the patient.

After 24 h of incubation at 37 °C, the culture plates were observed for bacterial growth and positive samples were selected for further laboratory processing. Culture plates without bacterial growth were re-incubated for another 24 h at 37 °C and observed for bacterial growth again. Negative culture samples were reported as negative for bacterial growth after 48 h, while the blood samples were incubated and checked for 7 consecutive days and re-inoculated after every 2 days to observe for any bacterial or fungal growth. The isolated organisms from the respiratory samples can be colonizers, but considering the immunocompromised status of patients, signs, and symptoms, cultures that were sent by clinicians suspecting bacterial infection (in COVID-19 positive patients) based on clinical signs and symptoms of the patients were included in the study [16,17].

After obtaining the growth, the morphology of each bacterial colony was observed and noted. Colonial morphology combined with Gram stain results and biochemical tests such as catalase, coagulase, oxidase, triple sugar iron (TSI), urease, analytical profile index (API) 20E, API 20NE, API *Staph*, and API *Strept* were employed to identify the organisms. A description of each biochemical test used has been given in the Table 1.

In the current study, the possibility of asymptomatic bacteriuria was kept in mind as the asymptomatic bacteriuria patients lacked the appearance of signs and symptoms that might be linked to the presence of bacteria in the urine. The assessment and treatment of symptomatic bacteriuria or urinary tract infections (UTIs) are very different from this, and urine cultures could be used to diagnose asymptomatic bacteriuria. A catheterized specimen or one that has been appropriately obtained using a clean-catch technique may be acceptable. The Infectious Diseases Society of America (IDSA) says that a bacterial infection is active when there are 10^5^ colony forming units (CFU)/mL of one type of bacteria in the urine. Furthermore, the presence of >4 pus cells also ruled out the possibility of neglecting the asymptomatic bacteriuria [15]. Furthermore, the presence of pathogenic bacteria in upper respiratory tract samples was defined as an upper respiratory tract infection, while the presence of pathogenic bacteria in lower respiratory tract samples was defined as lower respiratory tract infections, and the presence of bacteria in the blood samples was defined as bacteremia.

### 2.3. Antibiotic Susceptibility Testing (AST)

Isolated bacterial isolates were processed for AST by the disc diffusion method as per the recommendations by the Clinical Laboratory Standard Institute (CLSI) guidelines [18]. To perform the AST, MacFarland of 0.5 concentration was prepared using a density check technique for each of the isolates. This suspension was then spread on Muller Hinton (MH) agar plates and the antibiotics were placed on them according to CLSI guidelines, followed by incubation at 37 °C for 24 hrs. After the 24 h incubation, a zone of inhibition (ZOI) for each antibiotic was recorded as Resistant (R), Intermediate (I), and Sensitive (S). The ZOIs against the tested antibiotics were followed from the CLSI guidelines [15]. 

The tested antibiotics for Gram-positive isolates were amikacin, ampicillin, tetracycline, ciprofloxacin, tobramycin, gentamicin, teicoplanin, vancomycin, chloramphenicol, fusidic acid, linezolid, cefoxitin, clindamycin, erythromycin, nitrofurantoin, fosfomycin, azithromycin, levofloxacin, and cotrimoxazole-trimethoprim. Antibiotics used for Gram-negative isolates were amikacin, tobramycin, gentamicin, tetracycline, ciprofloxacin, levofloxacin, amoxicillin-clavunate, ceftriaxone, ceftazidime, cefepime, cefixime, cefuroxime, cotrimoxazole-trimethoprim, nitrofurantoin, fosfomycin, nalidixic acid, imipenem, meropenem, colistin, polymyxin b, minocycline, and piperacillin-tazobactam.

To validate the results of AST, American Type Culture Collections (ATCC) were used such as *E. coli* 25922, *Staphylococcus aureus* (*S. aureus*) 29213, *Enterococcus faecalis* (*E. faecalis*) 29212, and *Staphylococcus epidermidis* (*S. epidermidis*) 25923.

### 2.4. Statistical Analysis

All of the obtained data were entered into Microsoft Office Excel spreadsheet version 2016 (Microsoft, Washington, DC, USA), Minitab (Pennsylvania State University, USA), and SPSS version 26.0 (IBM, New York, NY, USA). After finishing with the data entry in data sheets, descriptive statistics were run on the data to obtain the absolute numbers (*n*) and was reported in percentages (%).

## 3. Results

### 3.1. Demographic Characteristics of Study Population

From the total 1251 studied patients, 1081 patients did not need oxygen, *n* = 74 were on ventilator, and *n* = 96 were oxygen dependent. The population of infected males was higher as compared to the female population in all three groups of the study, i.e., oxygen dependent (*n* = 38), oxygen independent only (*n* = 658), and on ventilator (*n* = 45). The highest number of COVID-19 infected patients were between the ages of 30 and 50 years. The characteristics of the COVID-19 patients with no oxygen support (NOS), oxygen support (OS), and ventilator dependent (VD), admitted to hospital are given in Table 2.

### 3.2. Sample-Wise Prevalence of Positive Bacterial Culture among COVD-19 Patients

Among all the collected samples, the highest positivity rate was recorded among tracheal aspirate which was 4.88%, followed by urine samples (4.56%). On the other hand, the lowest positivity rate was observed for pus samples followed by wound swabs (Table 3, Figure 1). The pus samples were collected using swabs from the deepest part of the wound avoiding the superficial microflora, and swabs were soaked in pus (if possible). The UTIs were seen in a total of 203 patients, while *n* = 157 were suffering from upper respiratory tract infection, *n* = 328 from lower respiratory tract infection, and *n* = 443 from bacteremia.

### 3.3. Antibiotic Susceptibility Testing

Out of 234 positive samples, only 39 were Gram-positive isolates, while the rest of the 195 were Gram-negative isolates. Among the Gram-negative isolates were *Klebsiella pneumonia, E. coli, A. baumannii, Pseudomonas aeruginosa* (*P. aeruginosa*)*, Proteus vulgaris*, *Klebsiella oxytoca* (*K. oxytoca*), *Enterobacter cloacae*, *Stenotrophomonas maltophilia*, *Citrobacter freundii*, and *Serratia liquefaciens.* Organisms with at least more than 10 isolates were processed for sensitivity.

*Enterococcus faecalis* was the most commonly isolated organism among the Gram-positive bacteria which accounted for 22 of all Gram-positive isolates. The antibiotic resistance pattern for Gram-positive isolates is given in Table 4.

Among 39 Gram-positive isolates, 22 were *E. faecalis*; 10 were *S. aureus*; while the remaining 7 isolates consisted of *S. epidermidis, Streptococcus pneumoniae* (*S. pneumoniae*)*,* and *Streptococcus spp.* Organisms with at least 10 isolates were included in the study.

Among the Gram-positive isolates, the highest resistance rate was noted against ciprofloxacin, while no bacterial isolate was found resistant to linezolid. Vancomycin was found to be as effective as linezolid against *S. aureus*, while some resistance (ranging from 1.2 to 5.2% among NOS, OS, and VD) was observed when tested against *E. faecalis*. Among the aminoglycosides group of antibiotics, amikacin was found to be the most effective antibiotic with a very low resistance rate (2.3 to 6.1%) among NOS, OS, and VD patients (Table 5).

Out of 195 Gram-negative isolates, *Proteus vulgaris* (*n* = 1), *K. oxytoca* (*n* = 4), *Enterobacter cloacae* (*n* = 2), *Stenotrophomonas maltophilia* (*n* = 2), *Citrobacter freundii* (*n* = 3), and *Serratia liquefaciens* (*n* = 1) were not included in the study because of a low number of isolates. 

Among Gram-negative isolates, *E. coli* was found most resistant to ampicillin (NOS = 98.2%, OS = 100%, and VD = 100%), followed by amoxicillin-clavunate (NOS = 88.7%, OS= 90.3%, and VD = 90.6%). The highest resistance was also observed for amoxicillin-clavunate, when tested against *K. pneumonia* (NOS = 90.0%, OS = 93.4%, and VD = 94.7%). Carbapenem resistance was slightly higher for *K. pneumonia* (NOS = 16.7%, OS = 17.2%, and VD = 18%) as compared to *E. coli* (NOS = 15.1%, OS = 15.6%, and VD = 16.7%) (Table 6).

*Acinetobacter baumannii* showed the most resistance against the cephalosporin group, and among cephalosporin, the highest resistance was observed against ceftriaxone (100%) among NOS, OS, and VD. The highest sensitivity of *A. baummanii* was observed when tested against colistin (100% sensitive among NOS, OS, and VD). *P. aeruginosa* isolates showed high susceptibility to carbapenems and aminoglycosides, while the lowest susceptibility was noted against fluroquinolones.

## 4. Discussion

Severe Acute Respiratory Syndrome SARS-CoV-2 due to the corona virus novel strain’s COVID-19 pandemic led to serious bacterial co-infections, especially in intensive care unit-admitted patients [5,19]. For the current study, 1251 COVID-19 positive patients were recruited, out of which most infected patients were male who were on life support oxygen (289) and 238 were on a ventilator. The highest positivity rate was observed in tracheal aspirate samples. Secondary infections were observed in 31% of ICU patients and 10% of all patients in a Wuhan investigation of 41 individuals [4]. 

A previous study from Wuhan indicated that 11/68 (16%) of 68 patients who died had secondary illnesses, while no further information was provided [20]. In the present study, out of 1251 positive COVID-19 patients, 234 (18.72%) were having a secondary bacterial infection. Another study in Italy found that of 16,654 critically ill patients who died of a SARS-CoV-2 infection, 11% had bacterial and fungal co-infections [21]. In a previously conducted study, *Streptococcus pneumoniae*, *S. aureus*, *K. pneumoniae*, *Haemophilus influenzae*, *Mycoplasma pneumoniae, A. baumannii, Legionella pneumophila,* and *Clamydia pneumoniae* were the most commonly detected co-pathogens of SARS-CoV-2 clearly showing an abundance of Gram-positive bacteria [22]. As compared to the current study, the prevalence of Gram-negative bacteria was more than Gram-positive, and the most common isolate was *E. coli.* Both Gram-positive isolates tested positive for resistance to ciprofloxacin, the most common antibiotic, whereas neither sample tested positive for linezolid resistance. For *E. faecalis*, vancomycin was found to be no less effective or no more effective than linezolid in terms of *S. aureus* resistance (range from 1.2 to 5.2% among NOS, OS, and VD). 

Results of the previous study from Lahore, Pakistan found that the most common bacterial infections among COVID-19 patients admitted in the SICU were caused by *E. coli* (32%) [9]. The *E. coli* was also the most commonly isolated bacteria among the hospitalized COVID-19 patients [5]. These results are consistent with the results of the current study, as in the current study the *E. coli* was the most prevalent organism detected in different samples from critically ill COVID-19 patients. However, the findings of Wolfe and colleagues are different from ours, as the most commonly isolated organism among Gram-negatives in their study was *K. pneumoniae* and among Gram-positives, *S. aureus* was the most commonly isolated [23]. In another study conducted by Chong et al. all the organisms were suspected to be causing co-infections, but they suspected them to be hospital acquired as these organisms (*A. baumannii*, *K. pneumoniae*, *E. coli*, etc.) are usually responsible for causing such infections [24].

The results of the current study showed that *A. baumannii* isolates were highly resistant to all tested antibiotics, which were similar to the results of a study conducted by Vijay et al. in which 47% of the infected patients were infected with MDR organisms, and among them *A. baummanii* showed the highest resistance against all tested antibiotics [25]. In the current study, resistance to amikacin has ranged from 2.3% to 6.1% among NOS, OS, and VD, making it the most effective aminoglycoside. Meanwhile, Gram-negative bacterial isolates from blood and urine were found to be highly resistant to amoxicillin/clavulanic acid, ampicillin, and erythromycin. 

MRSA and MSSA were among the Gram-positive pathogens that were isolated, and they were sensitive to vancomycin, teicoplanin, tigecycline, linezolid, and daptomycin, among other antibiotics in a previously published study [20]. The prevalence of MRSA in our study was 14.9%, which is comparable to the results of another study conducted by Hassan Mahmoudi et al. according to which it was 13.96% [15]. Another study reported an increase in the prevalence of MRSA from 3.53% to 25.30% in COVID-19 pathogens [26]. 

Most of *A. baumannii’s* resistance was found in the cephalosporin group, with the highest level of resistance found in ceftriaxone. When *A. baummanii* was tested against colistin, it was shown to be the most sensitive (100 percent sensitive among NOS, OS, and VD). *P. aeruginosa* isolates showed a high resistance rate against fluroquinolones. However, the most effective antibiotics were carbapenem and aminoglycosides. On the other hand, in another published research study, *P. aeruginosa* and *E. coli* were the bacteria that were most frequently isolated as multi-drug resistant (MDR) and related with hospital acquired superinfections [27]. A similar finding was also reported in another study, according to which *E. coli* was found to be highly resistant to trimethoprim/sulfamethoxazole and piperacillin/tazobactam [28].

## 5. Conclusions

Patients infected with SARS-CoV-2 are frequently found to have co-infections with a variety of bacteria, which have been shown to have a substantial impact on the severity and fatality rates of COVID-19. However, our understandings of co-infecting organisms, their interactions with one another, and their eventual interactions with the hosts are limited. Furthermore, viral co-infection promotes bacterial adhesion; disrupts the tight junction and epithelial barrier integrity, allowing bacteria to transmigrate across cells; and alters both innate and adaptive immune responses, making the lungs more vulnerable to SARS-CoV-2 infections in the future.

## Figures and Tables

**Figure 1 medicina-58-00904-f001:**
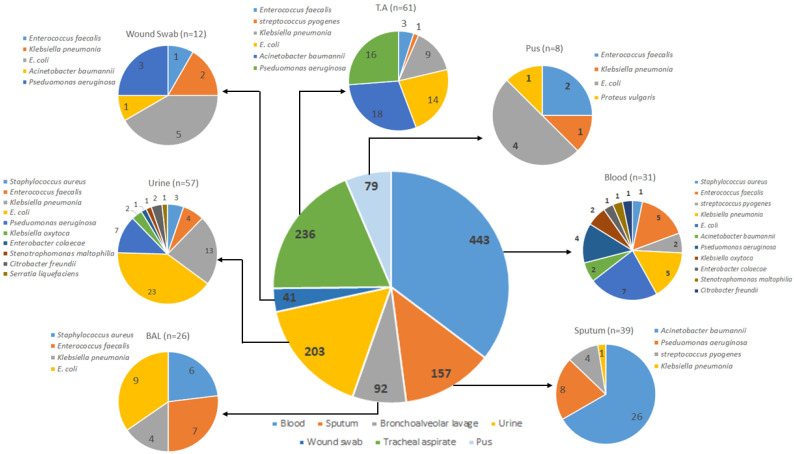
Sample-wise prevalence of bacterial isolates. *n* = number of samples. T.A: tracheal aspirate.

**Table 1 medicina-58-00904-t001:** Description of biochemical tests used.

Name of the Test	Description
Catalase	Used to differentiate between *Staphylococcus* spp. and *Streptococcus* spp.
Coagulase	Used to differentiate between *Staphylococcus aureus* and other *Staphylococcus* spp.
Oxidase	Used to identify *Pseudomonas* spp., *Burkhulderia* spp., and *Stenotrophomonas* spp.
Citrate, Indole, TSI, and Urease	Used to identify organisms based on their ability to utilize the substrates in each test.
API strips (API 20E, API 20NE, API staph, and API strept)	API strips comes with a range of biochemical tests that generate a code for each isolate and a database that identifies the organism using the code.

**Table 2 medicina-58-00904-t002:** Clinical and demographic characteristics of study population.

Variables		Characteristics of Study Population					
		No Oxygen Support (*n* = 1081)		Oxygen Support (*n* = 96)		Ventilator Dependent (*n* = 74)	
		f	%	f	%	f	%
Age	<30	331	30.62%	15	15.63%	13	17.57%
	30–50	527	48.75%	34	35.42%	27	36.49%
	>50	223	20.63%	47	48.96%	34	45.95%
Gender	Male	658	60.87%	56	58.33%	45	60.81%
	Female	423	39.13%	40	41.67%	29	39.19%
Underlying Disease	Diabetes mellitus	130	12.03%	11	11.96%	10	13.51%
	Hypertension	134	12.40%	10	10.87%	12	16.22%
	Kidney diseases	142	13.14%	25	27.17%	18	24.32%
	Gastrointestinal disorders	128	11.84%	14	15.22%	9	12.16%
	Liver disease	201	18.59%	17	18.48%	13	17.57%
	None	346	32.01%	19	19.79%	12	16.22%

f = frequency; % = percentage; *n* = number of samples/isolates/responses.

**Table 3 medicina-58-00904-t003:** Sample-wise prevalence of positive samples for pathogens, among COVD-19 patients.

Samples (*n* = 1251)	Positive Samples	
	f	%
Blood (*n* = 443)	31	2.48
Sputum (*n* = 157)	39	3.12
Bronchoalveolar lavage (*n* = 92)	26	2.08
Urine (*n* = 203)	57	4.56
Wound swab (*n* = 41)	12	0.96
Tracheal aspirate (*n* = 236)	61	4.88
Pus (*n* = 79)	08	0.61

f = frequency, % = percentage, *n* = number of samples/isolates/responses.

**Table 4 medicina-58-00904-t004:** Antibiotic resistance patterns among Gram-positive isolates.

Antibiotics	Percentage Resistance					
	*Staphylococcus aureus* (*n* = 10)			*Staphylococcus aureus* (*n* = 22)		
	NOS	OS	VD	NOS	OS	VD
Amikacin	2.3	4.8	6.1	NT	NT	NT
Chloramphenicol *	10.2	10.8	12.5	NT	NT	NT
Cefoxitin	14.4	14.6	15.7	NT	NT	NT
Ciprofloxacin	63.3	70.1	84.7	70.5	83.8	84.0
Co-trimoxazole	15.9	8.2	7.4	NT	NT	NT
Clindamycin	32.8	29.6	28.2	NT	NT	NT
Erythromycin *	70.4	50.5	46.3	55.7	70.1	79.8
Fusidic acid *	13.2	9.3	8.0	NT	NT	NT
Gentamicin	4.8	7.6	7.9	NT	NT	NT
Linezolid	0.0	0.0	0.0	NT	NT	NT
Penicillin	NT	NT	NT	10.4	12.0	14.1
Tetracycline	56.0	56.2	58.6	46.1	53.5	57.2
Tobramycin	15.0	15.4	17.1	NT	NT	NT
Levofloxacin	76.3	62.6	63.1	NT	NT	NT
Vancomycin	0.0	0.0	0.0	3.5	3.9	4.5

* Not included in urinary isolates. NOS: no oxygen support; OS: oxygen support; VD: ventilator dependent; *n* = number of samples/isolates/responses.

**Table 5 medicina-58-00904-t005:** Antibiotic resistance patterns among *Enterobacteriaceae*.

Antibiotics	Resistance Percentage (%)					
	*Escherichia coli* (*n* = 62)			*Klebsiella pneumoniae* (*n* = 35)		
	NOS	OS	VD	NOS	OS	VD
Ampicillin	98.2	100	100	NT	NT	NT
Amoxicillin clavunate	88.7	90.3	90.6	90.0	93.4	94.7
Amikacin	12.0	13.8	14.2	13.2	15.4	16.8
Ceftriaxone	42.1	54.5	55.3	64.1	68.9	70.0
Cefuroxime	57.3	67.1	70.2	79.0	83.2	85.6
Cefixime	57.3	67.1	70.2	77.1	84.5	88.4
Chloramphenicol *	48.1	54.5	59.8	67.5	68.9	70.5
Ciprofloxacin	52.8	58.2	58.6	87.6	90.8	93.6
Co-trimoxazole	24.2	27.3	30.9	30.2	33.3	35.1
Gentamicin	38.3	42.7	46.3	57.7	62.7	68.4
Fosfomycin **	11.2	14.3	15.8	NT	NT	NT
Imipenem	4	4.4	5.2	4.3	4.6	4.7
Meropenem	3.9	4.3	5.1	4.2	4.6	4.7
Nitrofurantoin **	13.5	15.6	17.2	42.5	56.4	59.0
Piperacillin-tazobactam	18.0	21.4	25.7	20.6	23.6	29.6
Tetracycline	14.7	16.8	22.3	45.2	52.4	56.7
Tobramycin	35.8	38.3	43.6	51.3	55.5	57.7
Colistin	0.0	0.0	0.0	0.0	0.0	0.0
Polymyxin B	0.0	0.0	0.0	0.0	0.0	0.0

* Not included in urinary isolates. ** Only reported in urinary isolates. NT: not tested; NOS: no oxygen support; OS: oxygen support; VD: ventilator dependent; *n* = number of samples/isolates/responses.

**Table 6 medicina-58-00904-t006:** Antibiotic resistance patterns among non-*Enterobacteriaceae*.

Antibiotics	Resistance Percentage (%)					
	*Acinetobacter baumannii* (*n* = 47)			*Pseudomonas aeruginosa* (*n* = 38)		
	NOS	OS	VD	NOS	OS	VD
Amikacin	33.5	34.8	37.5	12.2	12.8	14.6
Ceftazidime	82.3	85.1	86.8	44.6	55.7	56.4
Ciprofloxacin	78.1	84.3	85.7	81.12	88.2	88.6
Levofloxacin	72.6	81.4	84.8	NT	NT	NT
Co-trimoxazole	81.2	90.2	93.1	NT	NT	NT
Gentamicin	80.3	88.4	90.0	13.4	14.7	17.5
Imipenem	61.01	65.1	65.8	16.3	17.4	17.9
Meropenem	59.2	64.1	64.6	16.3	17.3	17.6
Piperacillin-tazobactam	68.0	78.5	84.8	17.2	18.6	19.0
Tetracycline	85.7	93.4	96.2	NT	NT	NT
Tigecycline	1.2	1.3	1.3	NT	NT	NT
Tobramycin	2.5	5.8	8.8	0.0	0.0	0.0
Colistin	0.0	0.0	0.0	NT	NT	NT
Polymyxin B	0.0	0.0	0.0	NT	NT	NT
Cefepime	83.2	83.5	85.7	42.4	54.2	57.9

NOS: no oxygen support; OS: oxygen support; VD: ventilator dependent; *n* = number of samples/isolates/responses.

## Data Availability

The data related to the current study will be available upon a reasonable request to the corresponding author.

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
