# Peer review of "Evaluation of Bi-Lateral Co-Infections and Antibiotic Resistance Rates among COVID-19 Patients in Lahore, Pakistan"

_medicina, 2022, doi:10.3390/medicina58070904_

Round 1

Reviewer 1 Report

I reviewed Azka Rizvi's work related to bacterial co-infections in critically ill COVID-19 patients admitted to a tertiary care hospital in Lahore, Pakistan.The work helps to clarify important aspects of this recent disease, providing an analysis of a specific local reality. However, some aspects need to be improved

In the title I would remove  the article: “identification and prevalence of bacterial coinfections and  antibiotic susceptibility patterns among critically Ill COVID-19 3 patients”

Authors need to better define study design and patients. For example, they must describe the capacity of the hospital where the study took place, how COVID-19 was diagnosed, the procedures used, the definitions relating to the clinical condition of the patients analyzed etc etc (I recommend being inspired by the study of ref 16 that they themselves cite). The same would be done for the part of "Data collection and outcomes" related to patients.

The legends of the figures and tables must be expanded so that the abbreviations used are explained (for example, the use of "f" must be explained, even if intuitive)

This sentence "Ciprofloxacin was found to be the most resistant antibiotic among both gram-positive isolates" (line 126) is profoundly wrong. It is the bacteria that are resistant to the drug and not the drug that is resistant to bacteria!

Others similar are:

-“ Among aminoglycosides, amikacin was found to be the most sensitive  drug with resistance…” line 130

- “ While fluroquinolones were the most resistant to P. aeruginosa “ line  191

- Line 150: it is necessary to correct in "were found". The actual sentence is "Carbapenems and aminoglycosides found to be most effective against P. aeruginosa ..."

- Line 148: there is probably a parenthesis that needs to be closed     (100%

-The name of the bacteria should be written the first time all in full, the second in shortened form (ie Escherichia coli and then E.coli) This rule has not always been respected in the work

I do not understand the statistical analysis that was made in the work. How were the results evaluated and compared with each other? Is there significance? (I also recommend for this part to be inspired by the study of ref 16 that the same authors cite).

The discussion section  absolutely needs to be broadened and must also be addressed by citing other similar works.

Author Response

Reviewer 1

Comments and Suggestions for Authors

I reviewed Azka Rizvi's work related to bacterial co-infections in critically ill COVID-19 patients admitted to a tertiary care hospital in Lahore, Pakistan. The work helps to clarify important aspects of this recent disease, providing an analysis of a specific local reality. However, some aspects need to be improved.

In the title I would remove the article: “identification and prevalence of bacterial coinfections and antibiotic susceptibility patterns among critically Ill COVID-19 3 patients”

Response: Dear reviewer, we would like to appreciate your suggestion for the change in article title. As per you suggestion we have revised the title of article.

Authors need to better define study design and patients. For example, they must describe the capacity of the hospital where the study took place, how COVID-19 was diagnosed, the procedures used, the definitions relating to the clinical condition of the patients analyzed etc etc (I recommend being inspired by the study of ref 16 that they themselves cite). The same would be done for the part of "Data collection and outcomes" related to patients.

Response: (Line: 105-110, 119-122, 129-132, 139-148) Dear reviewer, thank you for your valuable suggestions. The section about sample collection, isolation and identification of bacterial isolates has been revised according to the comments. Also, we want to mention that, the reason not to rule out the colonizers, and asymptomatic bacteriuria has been added in the revised manuscript. All the patients included in the study were PCR confirmed and clinically diagnosed. Official Govt. department called National Command and Operation Center of Pakistan (NCOC) dealt with all COVID-19 patient data and PCR results. Any patient with a positive PCR report was given an option (on the basis of severity of the disease) to either isolate themselves in homes or admit in public or private hospitals which were authorized as COVID-19 centers by NCOC. Furthermore, might be because of the typing or formatting error it may change, but the Ref. # 16, is a review article and don’t contain any relevant information for this respective section.

The legends of the figures and tables must be expanded so that the abbreviations used are explained (for example, the use of "f" must be explained, even if intuitive)

Response: Dear reviewer, thank you for your valuable comment. The legends for the tables and figure have been written in the revised manuscript.

This sentence "Ciprofloxacin was found to be the most resistant antibiotic among both gram-positive isolates" (line 126) is profoundly wrong. It is the bacteria that are resistant to the drug and not the drug that is resistant to bacteria!

Response: (Line: 186-187) Sentence has been rephrased.

Others similar are:

-“ Among aminoglycosides, amikacin was found to be the most sensitive  drug with resistance…” line 130

Response: (Line: 190-191) Sentence has been rephrased.

- “ While fluroquinolones were the most resistant to P. aeruginosa “ line  191

Response: (Line: 267) Sentence has been rephrased.

- Line 150: it is necessary to correct in "were found". The actual sentence is "Carbapenems and aminoglycosides found to be most effective against P. aeruginosa ..."

Response: (Line: 211-213) Sentence has been rephrased.

- Line 148: there is probably a parenthesis that needs to be closed     (100%

Response: (Line: 209) Corrected.

-The name of the bacteria should be written the first time all in full, the second in shortened form (ie Escherichia coli and then E. coli). This rule has not always been respected in the work

Response: Dear reviewer, thank you for your valuable comment. The names has been revised throughout the manuscript.

I do not understand the statistical analysis that was made in the work. How were the results evaluated and compared with each other? Is there significance? (I also recommend for this part to be inspired by the study of ref 16 that the same authors cite).

Response: (Line: 176-180) Dear reviewer, thank you for your valuable comment on statistical analysis. We have revised the section. Also, might be because of the typing or formatting error it may change, but the Ref. # 16, is a review article and don’t contain any relevant information for this respective section.

The discussion section absolutely needs to be broadened and must also be addressed by citing other similar works.

Response: (Line: 271-281, 289-292, 296) The discussion section has been thoroughly revised.

Reviewer 2 Report

Rizvi and colleague describe the frequency and type of bacterial isolates, together with antimicrobial susceptibility test, retrieved from clinical specimens of COVID-19 patients admitted to a tertiary care hospital in Lahore, Pakistan.

The manuscript requires a deep and extensive English editing.

In many parts of the manuscript, scientific soundness is totally absent.  

Introduction should be revised. Author should focus on the relationship between COVID-19 and secondary infections, the discussion about the hypercoagulable state is futile in this context.

Authors should clearly report which type of study they conducted (retrospective, observational).

Material and methods section should be placed before results.

The title doesn’t describe correctly what Authors report in the study. For each clinical samples (blood, BAL etc..), Authors simply report the frequency of positive cultures and the burden of different bacterial species, but we don’t know if these bacteria led to an infection or if they were colonizers.

This is especially true for tracheal aspirates, urine, and sputum (for examples, A. baumanii was isolated in 67% of tracheal aspirates, but it is well known that the clinical significance of this finding is questionable).

The title is, therefore, not correct.

The list of antibiotic susceptibility pattern of gram-positive and gram-negative bacteria is quite confusing, and the results are presented without a logical order, so that the reader is not able to take a clinical useful message from this study.

Author Response

Reviewer 2

Comments and Suggestions for Authors

Rizvi and colleague describe the frequency and type of bacterial isolates, together with antimicrobial susceptibility test, retrieved from clinical specimens of COVID-19 patients admitted to a tertiary care hospital in Lahore, Pakistan.

The manuscript requires a deep and extensive English editing. In many parts of the manuscript, scientific soundness is totally absent.  

Response: Dear reviewer, we would like to appreciate your kind efforts to the current manuscript. The manuscript has been thoroughly revised as per your comments and suggestions. Furthermore, the manuscript has been thoroughly revised for English proofreading and grammatical mistakes.

Introduction should be revised. Author should focus on the relationship between COVID-19 and secondary infections, the discussion about the hypercoagulable state is futile in this context.

Response: (Line: 82-94) The introduction section has been revised and the literature about secondary co-infection among COVID-19 patients has been added.

Authors should clearly report which type of study they conducted (retrospective, observational).

Response: (Line: 40) The “Observational study” has been written in the revised manuscript.

Material and methods section should be placed before results.

Response: Addressed

The title doesn’t describe correctly what Authors report in the study. For each clinical samples (blood, BAL etc..), Authors simply report the frequency of positive cultures and the burden of different bacterial species, but we don’t know if these bacteria led to an infection or if they were colonizers.

This is especially true for tracheal aspirates, urine, and sputum (for examples, A. baumanii was isolated in 67% of tracheal aspirates, but it is well known that the clinical significance of this finding is questionable).

The title is, therefore, not correct.

Response: Dear reviewer, thank you for your valuable suggestion. The title has been revised as per your comment. (Line: 119-122, 129-132, 139-148) Furthermore, the section about isolation and identification of bacterial isolates has been revised according to the comments. Also, the reason not to rule out the colonizers, and asymptomatic bacteriuria has been added in the revised manuscript.

The list of antibiotic susceptibility pattern of gram-positive and gram-negative bacteria is quite confusing, and the results are presented without a logical order, so that the reader is not able to take a clinical useful message from this study.

Response: (Table 3, 4 and 5) Dear reviewer, thank you for your valuable suggestion. The list of antibiotics in table 3, 4 and 5 has been updated in the revised manuscript. In this study samples were taken from a tertiary care hospital where patients were categorized as 1. No oxygen support required 2. Oxygen dependent and 3. Ventilator dependent. So, Antibiotic susceptibility pattern of gram positive and gram-negative isolates is also given according to these three categories discussed. Antibiotic susceptibility results are discussed in three table. In table 3 S. aureus and E. faecalis were discussed as they are both gram positive isolates in table 4 K. pneumonaie and E. coli were discussed as they are both Enterobacteriacae and in table 5 non-Enterobactericae are discussed.

Reviewer 3 Report

The authors provide insight into the prevalence of bacterial infections among COVID-19 patients in Pakistan. The article is well written in addition to the excellent use of tables and figures highlighting proportions of bacterial agents as well as their resistance patterns (strong methodology) This body of work will add to the ongoing surveillance of AMR in COVID-19. There are only minor modifications suggested:

1) Abstract Lines 41-43: "there is no enough data on the prevalence of bacterial co-infection in COVID-19 patients to reach the hypothesis." This is not necessarily true, Langford et al. from Toronto, Canada have a live website looking at rates of bacterial co-infections and secondary infections in patients with COVID-19. They have also published on this work - please amend and include appropriate references throughout the manuscript. 

Article: Bacterial co-infection and secondary infection in patients with COVID-19: a living rapid review and meta-analysis 

Article: Antibiotic prescribing in patients with COVID-19: rapid review and meta-analysis

2)What are the authors definition of secondary bacterial infection? How do we know if these patients were not co-infected? In the methods section please define as well as provide more detail regarding the standard protocol for COVID-19. Were these specimens collected from patients upon admission? Or later in their illness? This is important to tease out co-infection vs. secondary infection, especially for patients on O2 or ventilator support.

3) If possible, for Table 1, include A) time of patient admitted to hospital without oxygen, with oxygen and on ventilator support and B) time of isolation of pathogenic bacteria from admission for each category

4) In manuscript, please define Escherichia coli at first use, then use E. coli afterwards. In Figure 1, for consistency with the other species, please use Escherichia coli instead of E. coli.

5) A breakdown of the organisms identified in patients without oxygen support, with oxygen and with ventilator support would be useful (can include as supplemental material). This, in combination with time from admission to detection of organism can further elucidate use of empiric antibiotics during COVID and the isolation of more resistant organisms in those who are admitted for longer, especially since the authors are keen on describing bacterial infection as secondary infection.

Author Response

Reviewer 3

Comments and Suggestions for Authors

The authors provide insight into the prevalence of bacterial infections among COVID-19 patients in Pakistan. The article is well written in addition to the excellent use of tables and figures highlighting proportions of bacterial agents as well as their resistance patterns (strong methodology) This body of work will add to the ongoing surveillance of AMR in COVID-19. There are only minor modifications suggested:

1) Abstract Lines 41-43: "there is no enough data on the prevalence of bacterial co-infection in COVID-19 patients to reach the hypothesis." This is not necessarily true, Langford et al. from Toronto, Canada have a live website looking at rates of bacterial co-infections and secondary infections in patients with COVID-19. They have also published on this work - please amend and include appropriate references throughout the manuscript. 

Article: Bacterial co-infection and secondary infection in patients with COVID-19: a living rapid review and meta-analysis 

Article: Antibiotic prescribing in patients with COVID-19: rapid review and meta-analysis

Response: Dear reviewer, we would like to appreciate your efforts for the current manuscript and also would like to thank you for increasing our knowledge about the website by Langford et al, as we were not aware of this previously. Furthermore, to strengthen the literature of the current study, the suggested studies has been utilized and cited accordingly. Keeping in mind the updated data, the statement at line 41-43 of the previous version of manuscript has been removed from the revised manuscript.

2) What are the authors’ definition of secondary bacterial infection? How do we know if these patients were not co-infected? In the methods section please define as well as provide more detail regarding the standard protocol for COVID-19. Were these specimens collected from patients upon admission? Or later in their illness? This is important to tease out co-infection vs. secondary infection, especially for patients on O2 or ventilator support.

Response: (Line: 105-110) Dear reviewer, thank you for your valuable comment. We have added a statement in the sample collection section of material and methods in revised manuscript.

3) If possible, for Table 1, include A) time of patient admitted to hospital without oxygen, with oxygen and on ventilator support and B) time of isolation of pathogenic bacteria from admission for each category

Response: Dear reviewer, although it is a very good suggestion but due to certain managerial shortcomings, we are unable to make this change. However, we consider that this is very critical point raised by you so we will keep in mind during our next research project.

4) In manuscript, please define Escherichia coli at first use, then use E. coli afterwards. In Figure 1, for consistency with the other species, please use Escherichia coli instead of E. coli.

Response: (Line: 47) The Escherichia coli has been defined at its first place. Furthermore, to make it more clear in the figure 1, the legend for E. coli has been added.

5) A breakdown of the organisms identified in patients without oxygen support, with oxygen and with ventilator support would be useful (can include as supplemental material). This, in combination with time from admission to detection of organism can further elucidate use of empiric antibiotics during COVID and the isolation of more resistant organisms in those who are admitted for longer, especially since the authors are keen on describing bacterial infection as secondary infection.

Response: Dear reviewer, we really appreciate that this is again an excellent suggestion. However, since the current study was observational type, isolation of organisms based on clinical suspicion of secondary infection based on clinical symptoms only their antibiotic susceptibility testing was performed. This was done in order to help clinicians on selection of empirical therapy based on three categories (No Oxygen support, Oxygen Dependent and Ventilator Dependent) discussed in the study. Antibiotic susceptibility pattern for each category was discussed according to isolates to give an idea about their resistance pattern. We apologize that, at this stage because of data access issues and ethical consideration of the institution we are unable to re-access patient data for the above suggestion. We will really appreciate if you understand the situation of data access and ethical consideration from institution and allow us to let us proceed as in the current form.

Round 2

Reviewer 1 Report

The work has been improved but there are still many inaccuracies, especially of the English language.  The attached file reports these errors with the proposed changes 

Author Response

Reviewer 1

Comments and Suggestions for Authors

The work has been improved but there are still many inaccuracies, especially of the English language.  The attached file reports these errors with the proposed changes 

Response: Dear reviewer, we would like to appreciate your kind concerns about the current study. We really appreciate that your comments have improved the quality of the manuscript and it becomes better for the readers and scientific community. Furthermore, we have thoroughly revised the manuscript for English proofreading and grammatical mistakes.

Reviewer 2 Report

Reviewer 2

Comments and Suggestions for Authors

In green comments by the reviewer.

Rizvi and colleague describe the frequency and type of bacterial isolates, together with antimicrobial susceptibility test, retrieved from clinical specimens of COVID-19 patients admitted to a tertiary care hospital in Lahore, Pakistan.

The manuscript requires a deep and extensive English editing. In many parts of the manuscript, scientific soundness is totally absent.  

Response: Dear reviewer, we would like to appreciate your kind efforts to the current manuscript. The manuscript has been thoroughly revised as per your comments and suggestions. Furthermore, the manuscript has been thoroughly revised for English proofreading and grammatical mistakes.

The article now sounds better, and I acknowledge Authors Spelling efforts. However, pay atention to some spelling mistakes (ie line 152, 195...).

Introduction should be revised. Author should focus on the relationship between COVID-19 and secondary infections, the discussion about the hypercoagulable state is futile in this context.

Response: (Line: 82-94) The introduction section has been revised and the literature about secondary co-infection among COVID-19 patients has been added.

The introduction has been revised according to the suggestions.

Authors should clearly report which type of study they conducted (retrospective, observational).

Response: (Line: 40) The “Observational study” has been written in the revised manuscript.

Authors have correctly reported the study type. Please, report this at the end of the introduction section (ie, “this is a Retrospective observational study about... “

Material and methods section should be placed before results.

Response: Addressed

Authors have correctly revised the organization of the paper.

The title doesn’t describe correctly what Authors report in the study. For each clinical samples (blood, BAL etc..), Authors simply report the frequency of positive cultures and the burden of different bacterial species, but we don’t know if these bacteria led to an infection or if they were colonizers.

This is especially true for tracheal aspirates, urine, and sputum (for examples, A. baumanii was isolated in 67% of tracheal aspirates, but it is well known that the clinical significance of this finding is questionable).

The title is, therefore, not correct.

Response: Dear reviewer, thank you for your valuable suggestion. The title has been revised as per your comment. (Line: 119-122, 129-132, 139-148) Furthermore, the section about isolation and identification of bacterial isolates has been revised according to the comments. Also, the reason not to rule out the colonizers, and asymptomatic bacteriuria has been added in the revised manuscript.

The Authors changed the title and they included explanations about how they handled possibile airways and urinary tract colonizations. The explanations are quite “twisted” but, at least, the issue of “defining an Infection “ has been adress ed. For example, Authors could simply state how they defined a lower respiratory Infection, an urinary tract infections and so l’on…

There are still some problems. 1) samples “pus”, what does it mean? From wich body site the pus was cultured? 2) I would state how many infections were observed, not only the positivity rate of cultures. How many lower respiratory tract infections in total ? How many blood stream infections? How many UTI? Etc.. Please report this at the beginning of section 3.2.

In the title I would not mention “critically ill”, because Authors also report data about patients without oxygen support.

The list of antibiotic susceptibility pattern of gram-positive and gram-negative bacteria is quite confusing, and the results are presented without a logical order, so that the reader is not able to take a clinical useful message from this study.

Response: (Table 3, 4 and 5) Dear reviewer, thank you for your valuable suggestion. The list of antibiotics in table 3, 4 and 5 has been updated in the revised manuscript. In this study samples were taken from a tertiary care hospital where patients were categorized as 1. No oxygen support required 2. Oxygen dependent and 3. Ventilator dependent. So, Antibiotic susceptibility pattern of gram positive and gram-negative isolates is also given according to these three categories discussed. Antibiotic susceptibility results are discussed in three table. In table 3 S. aureus and E. faecalis were discussed as they are both gram positive isolates in table 4 K. pneumonaie and E. coli were discussed as they are both Enterobacteriacae and in table 5 non-Enterobactericae are discussed.

Tha tables are similar to the previous manuscript version, and I still think that they are not easy ti read. Foe example, it is not useful to report the e resistance rate of A. baumanii to ceftriaxone, because this bacteria is intrinsically resistant to ceftriaxone. However, in the text the Authors reported at least the most important findings.

Author Response

Reviewer 2

Comments and Suggestions for Authors

In green comments by the reviewer.

Rizvi and colleague describe the frequency and type of bacterial isolates, together with antimicrobial susceptibility test, retrieved from clinical specimens of COVID-19 patients admitted to a tertiary care hospital in Lahore, Pakistan.

The manuscript requires a deep and extensive English editing. In many parts of the manuscript, scientific soundness is totally absent.  

Response: Dear reviewer, we would like to appreciate your kind concerns about the current study. We really appreciate that your comments have improve the quality of manuscript and it becomes better for the readers and scientific community. Furthermore, we have thoroughly revised the manuscript for English proofreading and grammatical mistakes.

Response: Dear reviewer, we would like to appreciate your kind efforts to the current manuscript. The manuscript has been thoroughly revised as per your comments and suggestions. Furthermore, the manuscript has been thoroughly revised for English proofreading and grammatical mistakes.

The article now sounds better, and I acknowledge Authors Spelling efforts. However, pay atention to some spelling mistakes (ie line 152, 195...).

Response: Corrected.

Introduction should be revised. Author should focus on the relationship between COVID-19 and secondary infections, the discussion about the hypercoagulable state is futile in this context.

Response: (Line: 82-94) The introduction section has been revised and the literature about secondary co-infection among COVID-19 patients has been added.

The introduction has been revised according to the suggestions.

Authors should clearly report which type of study they conducted (retrospective, observational).

Response: (Line: 40) The “Observational study” has been written in the revised manuscript.

Authors have correctly reported the study type. Please, report this at the end of the introduction section (ie, “this is a Retrospective observational study about... “

Response: (Line: 99) Corrected.

Material and methods section should be placed before results.

Response: Addressed

Authors have correctly revised the organization of the paper.

The title doesn’t describe correctly what Authors report in the study. For each clinical samples (blood, BAL etc..), Authors simply report the frequency of positive cultures and the burden of different bacterial species, but we don’t know if these bacteria led to an infection or if they were colonizers.

This is especially true for tracheal aspirates, urine, and sputum (for examples, A. baumanii was isolated in 67% of tracheal aspirates, but it is well known that the clinical significance of this finding is questionable).

The title is, therefore, not correct.

Response: Dear reviewer, thank you for your valuable suggestion. The title has been revised as per your comment. (Line: 119-122, 129-132, 139-148) Furthermore, the section about isolation and identification of bacterial isolates has been revised according to the comments. Also, the reason not to rule out the colonizers, and asymptomatic bacteriuria has been added in the revised manuscript.

The Authors changed the title and they included explanations about how they handled possibile airways and urinary tract colonizations. The explanations are quite “twisted” but, at least, the issue of “defining an Infection “ has been adress ed. For example, Authors could simply state how they defined a lower respiratory Infection, an urinary tract infections and so l’on…

There are still some problems. 1) samples “pus”, what does it mean? From wich body site the pus was cultured? 2) I would state how many infections were observed, not only the positivity rate of cultures. How many lower respiratory tract infections in total ? How many blood stream infections? How many UTI? Etc.. Please report this at the beginning of section 3.2.

In the title I would not mention “critically ill”, because Authors also report data about patients without oxygen support.

Response: (Line 152-156, 193-197) Statement about pus, and the type of infections has been added in the revision.

The word “critically ill” has been removed from the title of manuscript.

The list of antibiotic susceptibility pattern of gram-positive and gram-negative bacteria is quite confusing, and the results are presented without a logical order, so that the reader is not able to take a clinical useful message from this study.

Response: (Table 3, 4 and 5) Dear reviewer, thank you for your valuable suggestion. The list of antibiotics in table 3, 4 and 5 has been updated in the revised manuscript. In this study samples were taken from a tertiary care hospital where patients were categorized as 1. No oxygen support required 2. Oxygen dependent and 3. Ventilator dependent. So, Antibiotic susceptibility pattern of gram positive and gram-negative isolates is also given according to these three categories discussed. Antibiotic susceptibility results are discussed in three table. In table 3 S. aureus and E. faecalis were discussed as they are both gram positive isolates in table 4 K. pneumonaie and E. coli were discussed as they are both Enterobacteriacae and in table 5 non-Enterobactericae are discussed.

Tha tables are similar to the previous manuscript version, and I still think that they are not easy ti read. Foe example, it is not useful to report the e resistance rate of A. baumanii to ceftriaxone, because this bacteria is intrinsically resistant to ceftriaxone. However, in the text the Authors reported at least the most important findings.

Response: Dear reviewer, we would like to appreciate your comments on the manuscript about the reporting of drug susceptibility testing. We have revised the reporting in table 3, 4 and 5 and also removed few antibiotics wherever necessary. Furthermore, the Ceftriaxone results for A. baumannii has also been removed. However, previously it was reported to double confirm the organism based on it’s intrinsic resistance.

This manuscript is a resubmission of an earlier submission. The following is a list of the peer review reports and author responses from that submission.